# Effects of a Subanesthetic Ketamine Infusion on Inflammatory and Behavioral Outcomes after Closed Head Injury in Rats

**DOI:** 10.3390/bioengineering10080941

**Published:** 2023-08-08

**Authors:** Haley F. Spencer, Martin Boese, Rina Y. Berman, Kennett D. Radford, Kwang H. Choi

**Affiliations:** 1Program in Neuroscience, Uniformed Services University, 4301 Jones Bridge Rd, Bethesda, MD 20814, USA; haley.spencer.ctr@usuhs.edu; 2Center for the Study of Traumatic Stress, Uniformed Services University, 4301 Jones Bridge Rd, Bethesda, MD 20814, USA; rina.berman.ctr@usuhs.edu; 3Daniel K. Inouye Graduate School of Nursing, Uniformed Services University, 4301 Jones Bridge Rd, Bethesda, MD 20814, USA; martin.boese@usuhs.edu (M.B.); kennett.d.radford.mil@health.mil (K.D.R.); 4Department of Psychiatry, Center for the Study of Traumatic Stress, Uniformed Services University, 4301 Jones Bridge Rd, Bethesda, MD 20814, USA

**Keywords:** mild traumatic brain injury, CHIMERA, ketamine, cytokines, axonal damage, rotarod

## Abstract

Traumatic brain injury (TBI) affects millions of people annually, and most cases are classified as mild TBI (mTBI). Ketamine is a potent trauma analgesic and anesthetic with anti-inflammatory properties. However, ketamine’s effects on post-mTBI outcomes are not well characterized. For the current study, we used the Closed-Head Impact Model of Engineered Rotational Acceleration (CHIMERA), which replicates the biomechanics of a closed-head impact with resulting free head movement. Adult male Sprague–Dawley rats sustained a single-session, repeated-impacts CHIMERA injury. An hour after the injury, rats received an intravenous ketamine infusion (0, 10, or 20 mg/kg, 2 h period), during which locomotor activity was monitored. Catheter blood samples were collected at 1, 3, 5, and 24 h after the CHIMERA injury for plasma cytokine assays. Behavioral assays were conducted on post-injury days (PID) 1 to 4 and included rotarod, locomotor activity, acoustic startle reflex (ASR), and pre-pulse inhibition (PPI). Brain tissue samples were collected at PID 4 and processed for GFAP (astrocytes), Iba-1 (microglia), and silver staining (axonal injury). Ketamine dose-dependently altered locomotor activity during the infusion and reduced KC/GRO, TNF-α, and IL-1β levels after the infusion. CHIMERA produced a delayed deficit in rotarod performance (PID 3) and significant axonal damage in the optic tract (PID 4), without significant changes in other behavioral or histological measures. Notably, subanesthetic doses of intravenous ketamine infusion after mTBI did not produce adverse effects on behavioral outcomes in PID 1–4 or neuroinflammation on PID 4. A further study is warranted to thoroughly investigate beneficial effects of IV ketamine on mTBI given multi-modal properties of ketamine in traumatic injury and stress.

## 1. Introduction

Traumatic brain injury (TBI) affects millions worldwide each year and constitutes a critical public health problem with no consensus on evidence-based treatments [1,2,3]. An estimated 80–90% of cases are classified as mild TBI (mTBI) [4,5], though the actual number of mTBI cases is likely underreported [6,7]. Though most mTBI patients fully recover within a few weeks, a subset do not. An estimated 20–30% of mTBI patients may suffer persistent adverse effects after a single brain injury [8,9]. These long-term consequences include physical, cognitive, emotional, social, and behavioral deficits [7]. Additionally, the adverse functional outcomes after mTBI are accompanied by physiological changes. The primary structural brain damage during mTBI is followed by a secondary injury caused by cascades of pathological and biochemical changes [10]. Targeted prevention of the secondary injury may have beneficial implications on long-term functional outcomes after mTBI.

Initial inflammation and subsequent chronic neuroinflammation are central mechanisms in developing behavioral and functional changes after secondary injury [11,12]. Inflammatory disruption after TBI includes changes in levels of inflammatory mediators, such as cytokines, both peripherally [13,14,15,16,17] and centrally [18,19], as well as an increase in glial cell activation and proliferation [12,20,21]. Therefore, understanding how inflammatory pathways are altered after injury may reveal therapeutic targets for treating mTBI [22] and preventing secondary injury. In turn, this could be used to prevent or treat some of the physical and cognitive consequences seen in mTBI patients after injury.

Ketamine, a multimodal dissociative anesthetic, is a common analgesic administered to combat-wounded service members in a forward-deployed setting [23,24,25]. It is used as a trauma analgesic and anesthetic alone or in conjunction with opioids [26] because it preserves respiration and hemodynamic stability after injury [23,27]. Despite the past dogma concerning increased intracranial pressure, ketamine is no longer contraindicated for TBI [28,29,30,31]. In fact, recent studies have shown that ketamine is safe to administer after injury and may provide neuroprotective benefits such as potent anti-inflammatory effects [28,32,33,34,35,36,37]. Anti-inflammatory properties of ketamine have been observed in cell culture [38,39], in rodent models of induced inflammation [40,41], and in some rodent TBI models [42,43]. Ketamine administration after TBI could, therefore, result in beneficial outcomes.

The present study used the Closed-Head Impact Model of Engineered Rotational Acceleration (CHIMERA) in laboratory rats. This novel and recently developed model produces a closed-head injury with resulting acceleration/deceleration movement. Compared to other preclinical models, CHIMERA produces an injury more translatable to clinical TBI mechanisms. This model can produce a mild TBI phenotype, similar to a human concussion [44]. Additionally, ketamine was given intravenously in this study, which is the principal administration route for humans following injury [23]. To our knowledge, no previous studies have investigated the effects of ketamine following CHIMERA injury. Therefore, we hypothesized that CHIMERA would produce a mild TBI phenotype and that administration of ketamine after injury would reduce inflammation and functional deficits post-injury.

## 2. Materials and Methods

### 2.1. Animals

Adult male Sprague–Dawley rats (9 weeks old upon arrival) were housed individually in clear Plexiglass cages in a climate-controlled environment. Animals were on a reversed 12 h light/dark cycle (lights off at 0600; lights on at 1800; testing during the dark cycle) with food and water available ad libitum. A jugular venous catheter (polyurethane; Instech, Plymouth Meeting, PA, USA) was surgically implanted under isoflurane anesthesia at Envigo Laboratories (Indianapolis, IN, USA) before animal arrival. The catheter was tunneled under the skin and connected to a vascular access button that exited the dorsal position between the rodent scapulae. Catheters were flushed twice weekly to verify catheter patency and locked with 0.1 mL sterile heparin/glycerol solution (1:1 dilution). Animals were handled before the experiment. All procedures were performed per the National Institutes of Health *Guide for the Care and Use of Laboratory Animals* and were approved by the Institutional Animal Care and Use Committee (IACUC) at the Uniformed Services University, Bethesda, MD, USA. Animals were randomly assigned to one of the six groups (injury condition-ketamine dose): Sham-0 (n = 13), Sham-10 (n = 14), Sham-20 (n = 16), CHIMERA-0 (n = 22), CHIMERA-10 (n = 23), and CHIMERA-20 (n = 22). Group numbers varied slightly to account for animals that lost catheter patency and did not give blood. The experimental timeline is shown in Figure 1.

### 2.2. CHIMERA Injury

Before the CHIMERA procedure, rats were anesthetized with isoflurane (4% for induction, 5 min, and 3% for maintenance via nose cone in the CHIMERA device, 1 min) mixed with 100% oxygen. Each animal was placed in a dorsal position in the CHIMERA device with adhesive straps holding the body in place. The head was centered over crosshairs on an aluminum plate, aligning the impact piston around the bregma area [45]. Nose cone isoflurane anesthesia was removed immediately before impact. A hole in the plate allowed a 200 g piston [46] to impact the head (1.5 J, 5.5 m/s velocity). Animals were given three impacts 5–10 s apart, allowing time to realign the head before each impact. After the injury, all rats received 150 mL acetaminophen (1 mg/mL) diluted in their drinking water for pain management for one day. Sham animals underwent all procedures except for impacts.

### 2.3. Ketamine Infusion

Animals received a ketamine or saline infusion one hour following the CHIMERA procedure. As previously described [47], racemic (±) ketamine hydrochloride (100 mg/mL) (Covetrus, Dublin, OH, USA) was diluted in 0.9% sterile saline before the infusion. The ketamine infusion (0, 10, or 20 mg/kg, IV) took place over two hours. Animals were given a loading dose of IV bolus ketamine or saline before being put into the infusion chambers (Med Associates Inc., St. Albans, VT, USA). Each chamber (14” L × 12” W × 15” H) was equipped with an infusion pump (Harvard Pump 11 Elite, Holliston, MA, USA) that used a 5 mL plastic syringe connected to a fluid swivel (Instech, Plymouth Meeting, MA, USA) by polyurethane tubing. This tubing was encased in a metal spring-wire tether that was magnetically attached to the metal cannula on the exit port of the catheter between the rat scapulae. The animals were allowed free movement in the chambers during the infusion due to the tether. Chambers contained two infrared photobeams to measure the spontaneous horizontal activity of animals throughout the infusion for locomotor activity levels, whereby a computer recorded the number of photobeam breaks.

#### Blood Sampling

Sample sizes per group vary; not all cytokines were detectable in all samples. Because cytokines peak within the first day after an injury [12], we collected plasma samples within 24 h post-injury. Blood (0.25 mL) was drawn from catheters one hour post-injury (immediately before the infusion), three hours post-injury (0 h post-infusion), five hours post-injury (2 h post-infusion), and 24 h post-injury. The timeline of blood sampling is shown in Figure 1. One hour before the first blood sample, catheters were flushed with saline to facilitate blood sampling. Blood sampling was minimally stressful and took about 15 s per animal. Blood was collected in microcentrifuge tubes containing 2.5 μL 0.5 M EDTA and centrifuged at 4000 rpm for 10 min at 4 °C. Plasma was aliquoted into tubes and frozen at −70 °C.

### 2.4. Cytokine Multiplex Immunoassay

An electrochemiluminescent multiplex immunoassay (Pro-inflammatory Rat Panel 2 V-PLEX, Meso Scale Discovery (MSD), Rockville, MD, USA) was used to quantify plasma inflammatory cytokine levels. The kit measures inflammatory cytokines interferon (IFN)-γ, interleukin (IL)-1β, IL-4, IL-5, IL-6, KC/GRO, IL-10, IL-13, and tumor necrosis factor (TNF)-α levels. As previously described [47], the MSD 96-well plate contained primary capture antibody for each cytokine and was blocked for one hour; then, standards and 4× diluted samples (50 μL per well) were added. The plate was then incubated at room temperature for two hours; all shaking was at 500 rpm. Between each step, the plate was washed with wash buffer (10× PBS with 0.05% Tween 20). Next, secondary detection antibodies for each cytokine were added, and the plate was incubated for two hours at room temperature. After washing, read buffer was added, and the plate was then read on the MSD Sector Image 6000 (Meso Scale Discovery, Rockville, MD, USA) utilizing the corresponding Workbench software. For each cytokine antibody, the lower limit of detection was 0.65 pg/mL (IFN-γ), 6.92 pg/mL (IL-1β), 0.69 pg/mL (IL-4), 14.1 pg/mL (IL-5), 13.8 pg/mL (IL-6), 1.04 pg/mL (KC/GRO), 16.4 pg/mL (IL-10), 1.97 pg/mL (IL-13), and 0.72 pg/mL (TNF-α). Due to catheter patency issues for blood draw and low detection limits for some cytokines, final sample sizes vary across cytokines in the time course.

### 2.5. Behavioral Tests

All behavioral apparatuses were cleaned with 70% ethanol between subjects and the Sani-cloth at the end of the testing period.

#### 2.5.1. Rotarod

Rotarod tests were performed according to the Preclinical Behavior and Modeling Core standard protocols at the Uniformed Services University. The Rotor-Rod system (San Diego Instruments, San Diego, CA, USA) with corresponding software was used. Photobeam breaks recorded latency to fall. Animals were placed in the rotarod room to habituate for one hour before testing with the lights off. Rats were trained on the rotarod one day before baseline testing. Training took place on a rod accelerating from 0 to 5 rpm for 30 s and then constant speed at 5 rpm for 60 more seconds. Testing on days one and three post-CHIMERA occurred with the rod accelerating from 0 to 40 rpm over 300 s. The maximum time on the rod was 90 s for training and 300 s for testing. Two rats were on the rod in each trial, with an empty lane between them. There were three trials per session, with 15 min rest periods between trials.

#### 2.5.2. Locomotor Activity

Locomotor activity was assessed days one and three post-CHIMERA in 40 × 40 × 36 cm SuperFlex Open Field boxes equipped with 16 by 16 grid of infrared photo beams tracking horizontal and vertical movement, connected to a computer using Fusion version 6.4M SuperFlex Edition software (Omnitech Electronics, Inc., Columbus, OH, USA). After one hour of acclimation in the dark testing room with a white noise machine turned on low (57.4 dB), rats were placed inside the boxes for one hour of locomotor activity assessment. The software recorded horizontal and vertical (rearing) activity.

#### 2.5.3. ASR/PPI

For acoustic startle reflex/pre-pulse inhibition (ASR/PPI) testing on days two and four post-CHIMERA, rats were first habituated in their home cages in a dark, quiet adjacent behavior room for one hour. Animals were previously acclimated to the acoustic startle chambers and animal holders. Four acoustic startle chambers (Med Associates Inc., product number: MED-ASR-PRO1) with animal holders for rats (Med Associates Inc., product number: ENV-262C) inside sound-attenuating cubicles were used and connected to a computer equipped with Startle Reflex software version 6.15 (Med Associates Inc, Fairfax, VT, USA). Chambers were calibrated using the average weight of animals (approximately 350 g). The ASR/PPI procedure had an initial three-minute acclimation period followed by 48 trials consisting of a mix of null stimulus, pre-pulse stimulus alone, pre-pulse plus startle stimulus, and startle stimulus alone, for a total of approximately 20 min. After the testing, rats were removed from the chambers and returned to their home cages. Percent PPI was calculated using the formula: $$\%\mathrm{PPI}=\frac{\mathrm{startle}\;\mathrm{stimulus}\;\mathrm{alone}\;-\left(\mathrm{pre}\text{-}\mathrm{pulse}\;\mathrm{plus}\;\mathrm{startle}\;\mathrm{stimulus}\right)}{\mathrm{startle}\;\mathrm{stimulus}\;\mathrm{alone}}\times 100$$

### 2.6. Brain Tissue Collection

Four days after injury, rats were deeply anesthetized with isoflurane and, once unresponsive to paw pinch, a midline thoracotomy was performed to expose the heart. An 18-gauge blunt-tip needle connected to a tube was inserted into the left ventricle and the right atrium was incised. After an initial washout with ~50 mL of ice-cold 0.1 M phosphate buffered saline (PBS), perfusion with ~100 mL ice-cold 10% neutral buffered formalin was delivered through a peristaltic perfusion pump until adequate fixation was reached. The brain was removed from the calvarium, post-fixed in 10% neutral buffered formalin for 24 h, and cryoprotected with 20% sucrose in PBS for 72 h before being frozen in dry ice and stored at −70 °C for histology experiments. Brains were then sectioned with a sliding Leica microtome (Lecia Biosystems, Nußloch, Germany; Richmond, IL, USA) and sections (40 μm) were stored in cryoprotectant solution at −20 °C until processing.

### 2.7. Histology

The Sham-0 and CHIMERA-0 groups were used for histology to determine whether CHIMERA injury induced pathological changes. Seven animals from each group for Iba-1 and GFAP staining and ten animals from each group for silver stain were randomly selected for processing. Ten animals from both the Sham-10 and CHIMERA-10 groups were also included for silver stain.

#### 2.7.1. Immunohistochemistry

Immunohistochemistry was performed according to a standard protocol [48,49]. After being in cryoprotectant for at least a week, free-floating sections were washed three times in tris-buffered saline with 0.05% triton (TBST), processed with 0.3% hydrogen peroxide for 30 min, rewashed, and then incubated with blocking buffer (TBST with 0.2% triton, 10% goat serum (Vector Laboratories, Burlingame, CA, USA, S-1000), and 0.02% bovine serum albumin (Fisher Scientific, Waltham, MA, USA, Fraction V)) for one hour shaking at room temperature. Primary antibodies of either glial fibrillary acidic protein (GFAP) for astrocytic marking (Epredia, Kalamazoo, MI, USA, REF: MS-1376-P0) or ionized calcium-binding adaptor molecule 1 (Iba-1) for microglial marking (Fujifilm Wako, Osaka, Japan, REF: 019-19741) were diluted 1:1000 in blocking buffer and incubated overnight at 4 °C. The following day, sections were washed and incubated in secondary antibody (biotin-SP-conjugated goat anti-mouse IgG for GFAP and biotin-SP-conjugated goat anti-rabbit for Iba-1, Jackson Immunoresearch, West Grove, PA, USA) diluted 1:300 in blocking buffer for one hour shaking at room temperature. After washing, sections were then incubated in ABC solution (Vectastain ABC Peroxidase kit, Vector Laboratories, Burlingame, CA, USA, REF: PK-4000) for 45 min, washed again, and developed with DAB Peroxidase Substrate kit (with nickel), 3,3′-diaminobenzidine staining solution (Vector Laboratories, Burlingame, CA, USA, REF: SK-4100) for three minutes. Sections were put in phosphate buffer (PB) to stop the DAB reaction and mounted onto glass slides to dry overnight. Slides were then dehydrated in ethanol gradients (75–100%), cleared in xylene, and cover slipped with Permount mounting medium (Fisher Chemical, Waltham, MA, USA).

#### 2.7.2. Silver Staining

Silver staining was performed following the FD NeuroSilver Kit II (FD Neurotechnologies, Columbia, MD, USA) protocol. Briefly, free-floating sections were stored in formalin for one week, then washed in ddH_2_O twice before incubating in each mixture of reagents (A, B, C, D, E, F, and G). After final incubation in G, sections were mounted on slides and dried overnight. The next day, they were cleared in xylene before being cover slipped with Permount mounting medium (Fisher Chemical, Waltham, MA, USA).

#### 2.7.3. Microscope Imaging and Quantification

Brain regions investigated for staining included the medial prefrontal cortex (mPFC), the corpus callosum (CC), the cerebellum (CB), and the optic tract (OPT). Images were taken on a Motic BA310 microscope (Motic, Kowloon, Hong Kong, China) with corresponding camera software at 4× magnification. To quantify regions of interest (ROIs) for Iba-1 and GFAP, the percent area stained was measured in black/white images using the Yen automatic thresholding method on Image J software (version 1.53t) [49]. Right and left values were averaged for the optic tract and the mPFC, and white and grey matter in the cerebellum were quantified separately to maintain consistent ROIs. The experimenter responsible for imaging and quantification was blinded to the injury condition.

For silver stain analysis, brain regions were imaged and scored for the intensity of silver stain on a scale from 0 to 3. A score of 0 indicated an absence of silver uptake, and scores of 1–3 indicated the presence of silver uptake, with 1 indicating minor uptake, 2 indicating moderate uptake, and 3 indicating extensive silver uptake, as measured by the prevalence of degenerating nerve fibers. Brain areas were scored by two experimenters blind to the injury condition, and scores were averaged for data analysis.

### 2.8. Statistical Analysis

Body weights, locomotor activity during the infusion, cytokine data after infusion, behavioral data, and brain tissue analysis are presented as mean ± standard error of the mean (SEM). Cytokine data after the infusion and all behavioral data were analyzed in SPSS (SPSS Software, version 28.0.1). Outliers were removed using the Grubbs test. For these data, three-way interaction mixed-model analysis of variance (ANOVA) (CHIMERA × ketamine × time) with time as a repeated measure was used, and Sidak tests were used as post hoc analyses. Body weight data were analyzed as a two-way repeated-measures ANOVA within GraphPad Prism (GraphPad Software, version 9.0). Cytokine data at one-hour post-CHIMERA and Iba-1 and GFAP brain histology (% area stained) were analyzed using unpaired *t*-tests within GraphPad Prism (GraphPad Software, version 9.0). Silver stain histology (scored on a scale of 0–3) was analyzed with Kruskal–Wallis non-parametric statistics and Dunn’s multiple comparisons. Cytokine data one-hour post-CHIMERA are shown by individual data points and the median lines. Brain tissue analysis is shown by mean ± SEM. All data were plotted using GraphPad Prism. Significance was determined through *p* < 0.05.

## 3. Results

### 3.1. Body Weight Changes following CHIMERA Injury

The body weights of animals were monitored daily from baseline to post-injury day 4 (Figure 2A). There were no significant differences in body weights between the groups on any day, indicating neither CHIMERA nor ketamine had any effect on body weight.

### 3.2. Ketamine-Induced Locomotor Activity Changes

Spontaneous locomotor activity was monitored during the two-hour IV ketamine infusion (0, 10, or 20 mg/kg) using infrared photo beams installed in each chamber. Due to dose- and time-dependent effects of ketamine on activity, data are plotted in the first and second hour separately (Figure 2B). Three-way interaction analysis revealed a significant main effect of ketamine (F_2,104_ = 11.308, *p* < 0.001) and time (F_1,104.019_ = 39.246, *p* < 0.001) but not CHIMERA (F_1,104_ = 0.797, *p* = 0.374). There was also a significant interaction effect of ketamine × time (F_2,104.019_ = 15.683, *p* < 0.001). Post hoc tests revealed that 10 mg/kg ketamine significantly reduced activity during the first hour compared to both saline (*p* < 0.001) and 20 mg/kg ketamine (*p* = 0.005). In contrast, during the second hour, 20 mg/kg ketamine significantly increased activity compared to saline (*p* = 0.002) and 10 mg/kg ketamine (*p* < 0.001). Activity in the second hour was reduced in both the Sham-0 and CHIMERA-0 groups (*p* < 0.0001) and in the Sham-10 group (*p* = 0.014). The reduced activity is likely due to habituation to the chamber after the first hour. These findings support the dose-dependent effects of ketamine on behavior (sedation vs. dissociation at lower and higher doses, respectively).

### 3.3. Time Course of Plasma Cytokines

The time courses of inflammatory cytokines in plasma samples were assayed following CHIMERA exposure and ketamine or saline infusion. The one-hour samples were collected immediately before the ketamine or saline infusion, so these data are shown in sham vs. CHIMERA groups with median bars (Figure 3). Unpaired *t*-tests showed no significant differences between the sham and CHIMERA groups.

The time course data are shown in their injury condition-dose groups and presented as time post-infusion (Figure 4). A three-way interaction analysis (CHIMERA × ketamine × time) revealed a significant main effect of ketamine on KC/GRO (F_2,57.843_ = 7.651, *p* = 0.001). Post hoc tests revealed that 10 mg/kg ketamine at 0 h (*p* = 0.002), 2 h (*p* = 0.008), and 24 h (*p* = 0.002) and 20 mg/kg at 0 h (*p* = 0.012), 2 h (*p* = 0.018), and 24 h (*p* = 0.018) reduced KC/GRO concentration compared to saline (Figure 4A). Three-way interaction analysis revealed significant main effects of ketamine (F_2,91.891_ = 5.055, *p* = 0.008) and time (F_2,181.186_ = 4.877, *p* = 0.009) on TNF-α. The 10 mg/kg ketamine dose significantly reduced TNF-α compared to saline immediately after the infusion (*p* = 0.041) (Figure 4B). Three-way interaction analyses revealed no significant effect of CHIMERA or ketamine but significant main effects of time on IL-10 (F_2,164.844_ = 10.722, *p* < 0.001), as shown in Figure 4C, IL-4 (F_2,137.646_ = 8.392, *p* < 0.001), IL-5 (F_2,81.037_ = 4.205, *p* = 0.018), and IL-6 (F_2,83.891_ = 3.237, *p* = 0.044). Additionally, there was a significant interaction between ketamine and time (F_4,46.167_ = 3.564, *p* = 0.013) on IL-1β. Post hoc tests indicated dose-dependent effects, with 20 mg/kg ketamine significantly reducing IL-1β compared to 10 kg/mg ketamine at 0 h (*p* = 0.041) (Figure 4D). These findings show the importance of investigating temporal changes in cytokines after injury or treatment.

### 3.4. Behavioral Outcome Measures

There were no differences between the groups at baseline on any behavioral measures. CHIMERA animals showed a significant reduction in their latency to fall off the rotarod compared to sham animals on day 3 post-injury (Figure 5). Three-way interaction analysis revealed significant main effects of CHIMERA (F_1,92_ = 5.158, *p* = 0.025) and time (F_1,90.478_ = 11.741, *p* < 0.001) on rotarod latency. Post hoc tests showed a significant difference between sham and CHIMERA groups on day 3 (*p* = 0.013) and approached significance on day 1 (*p* = 0.087). Thus, motor deficits following CHIMERA may be detectable at a later point rather than an earlier time point post-injury in this paradigm.

There were no significant differences in the open-field locomotor activity on post-injury days 1 and 3 between CHIMERA and sham animals. There were no significant effects in vertical activity (rearing) levels (Figure 6B). As expected, there was a significant main effect of time (F_1,103.021_ = 8.639, *p* = 0.004) on horizontal activity (Figure 6A) due to exploratory behavior (day 1) and habituation (day 3). These data suggest that the CHIMERA injury produces motor co-ordination deficits while producing no impairment of general activity levels.

Sensorimotor function was assessed using the ASR/PPI paradigm on post-injury days 2 and 4. Three-way interaction analyses revealed no significant main effects or interaction between CHIMERA, ketamine, and time on the level of ASR (100 and 110 dB) or PPI (100 and 110 dB), as shown in Figure 6C–F. As expected, animals showed greater startle response to the louder auditory pulse (110 dB) than to the 100 dB pulse, indicating no deficits in auditory function following CHIMERA injury.

### 3.5. Neuroinflammation and Axonal Damage

Brain tissue was analyzed to determine neuroinflammation (microglial and astrocyte activation) and axonal injury (white matter damage) on post-injury day 4. In all four brain areas selected (mPFC, CC, OPT, and CB), there were no significant differences between sham and CHIMERA groups in either microglial activation as seen with Iba-1 staining (Figure 7) or astrocyte activation as seen with GFAP staining (Figure 8). The optic tract showed axonal damage as seen with silver staining in both CHIMERA groups (*p* < 0.0001), but this was not significant in other brain areas, and there was no effect of ketamine (Figure 9). In the optic tract, multiple comparisons showed significant differences between the Sham-0 and CHIMERA-0 groups (*p* = 0.03), the Sham-0 and CHIMERA-10 groups (*p* = 0.004), the CHIMERA-0 and Sham-10 groups (*p* = 0.003), and the Sham-10 and CHIMERA-10 groups (*p* = 0.0003). These data indicate that the CHIMERA method did not produce detectable changes in neuroinflammation 4 days after injury but did induce axonal damage in a susceptible white matter tract, the optic tract.

## 4. Discussion

This study indicates that a single-session repetitive CHIMERA injury produced mTBI phenotypes. There were no significant changes in body weights or most behavioral measures except for motor co-ordination deficits seen in rotarod testing and only minor histological damage in the optic tract. This mild injury effect is consistent with most clinical mTBI studies. Ketamine significantly attenuated cytokine responses after infusion, despite the mild effects of CHIMERA injury on acute peripheral cytokine concentrations. Importantly, ketamine produced dose-dependent effects on several outcome measures, indicating the importance of investigating multiple drug doses in translational studies.

Ketamine produced dose-dependent effects on locomotor activity during the infusion, regardless of injury condition. The 10 mg/kg ketamine dose decreased activity in both sham and CHIMERA animals, while the 20 mg/kg dose showed stimulatory effects. This is consistent with our previous work showing that 10 mg/kg ketamine is slightly sedative [47]. Interestingly, the effects of ketamine persist over time, yet the animals receiving saline showed decreased locomotor activity in the second hour, likely due to habituation to the environment. The 20 mg/kg ketamine dose induced consistently high locomotor activity levels in both the first and second hours, revealing that this higher dose stimulates locomotor activity compared to saline. We likely did not see habituation in the 20 mg/kg ketamine groups due to the dissociative effects produced by a higher dose of ketamine, which is also consistent with our previous studies [50].

The dose-dependent effects of ketamine are important for the clinical translation of this model. Our previous work has shown that 10 mg/kg ketamine infused intravenously over two hours is analgesic without dissociation [50]. This behavioral profile is clinically relevant because subanesthetic doses are recommended for analgesic use in combat trauma [51]. As such, a behaviorally favorable dose that still provides immunomodulatory effects is ideal [36]. As the current data and previous data from our lab have shown [47], a 10 mg/kg ketamine dose has immunomodulatory effects on inflammatory cytokines and, therefore, is an important dose in preclinical studies investigating ketamine’s effects on immune mediators after injury.

Cytokine analysis of blood samples collected one hour after CHIMERA showed no significant differences between sham and injured animals. Both pro- and anti-inflammatory cytokines in the blood have been used as biomarkers of TBI and are associated with neuroimaging [13,52] and functional impairments in clinical studies [16,53]. The inconsistency in our dataset may be due to the temporal profile of cytokine concentrations after injury; one hour post-injury could be too early to observe cytokine changes in plasma in mTBI. However, as cytokines and chemokines are some of the first responders in an inflammatory insult, their concentrations likely peak on the first day after injury [12]. Additionally, peripheral cytokine concentration is highly influenced by the severity of TBI, as this directly impacts the level of disruption of the blood–brain barrier (BBB) [54]. The current model produces a mild TBI phenotype; therefore, it is unsurprising that there were no significant differences in peripheral cytokines one hour after injury. The time points selected for cytokine analysis may be critical for understanding a cytokine’s temporal profile that fluctuates over time regardless of injury and ketamine. Therefore, more studies would benefit from repeated blood sampling and a time course analysis as shown in this study.

A few proinflammatory cytokines often studied in TBI include IL-6, TNF-α, and IL-1β, which are consistently upregulated after injury [12]. CHIMERA did not significantly upregulate these cytokines in plasma at the selected time points in this study, which is likely a reflection of the mild nature of the injury paradigm. To our knowledge, this was the first study to investigate peripheral cytokines after CHIMERA injury. Previous studies have shown that all three of these cytokines are upregulated in brain tissue at various time points post-CHIMERA [55]. As the parameters of CHIMERA injury range across studies, one possible explanation for this finding could be that a more severe injury parameter is necessary to produce these results. Additionally, cytokine changes in the brain may not always correlate precisely with those seen in peripheral markers [19]. It should be noted that, in the present study, ketamine significantly reduced both TNF-α and IL-1β at the selected time points. A previous study showed that ketamine reduced IL-6 at a more delayed time than other cytokines [47], which could explain why reduction in IL-6 was not seen in this study. Therefore, with a TBI model that induced upregulation of these cytokines, we would expect that ketamine would reduce levels of those cytokines. This has been shown in other preclinical studies [32,56], and we would expect that future studies utilizing a CHIMERA injury resulting in the upregulation of those cytokines would show similar results.

The chemokine KC/GRO, the rodent analog of CXCL1, is a neutrophil attractant. Interestingly, not many studies have examined the role of CXCL1 in TBI, which is surprising as the infiltration of neutrophils after TBI is involved with BBB breakdown and can correlate with injury severity [12]. KC/GRO has been reported to peak four hours after controlled cortical impact in brain tissue in rats, which aligns with the timing of neutrophil infiltration [18]. In the current study, plasma KC/GRO levels approached a significant increase after CHIMERA. As expected, both doses of ketamine significantly reduced KC/GRO levels in both sham and CHIMERA animals, which has important implications for clinical use. As BBB breakdown and neutrophil infiltration are heavily involved in secondary injury after TBI, future studies should focus on this important chemokine, its temporal profile after injury, and how a drug such as ketamine may attenuate its response.

In the injury paradigm in the current study, animals that had sustained CHIMERA impacts showed significant deficits on the rotarod test on day 3 after injury. Similar motor impairments have been seen in mice up to one week after CHIMERA injury [45]. Interestingly, we did not see significant differences between the CHIMERA and sham animals on day 1 post-injury in this study, though the data seem to trend with the deficits seen in CHIMERA animals on day 3. However, rotarod deficits in this model are not always consistent, and the parameters of the CHIMERA injury may produce variable results [55]. Additionally, the timeline of deficits varies slightly across studies. Therefore, it is reasonable to assume that our CHIMERA method may produce motor deficits that are not detectable until a few days after injury.

Indeed, there was no difference in the spontaneous activity levels of animals on days 1 or 3 after injury, indicating that the current CHIMERA injury does not significantly impair overall motor function. Feng and colleagues similarly showed no differences between injured and sham animals in spontaneous activity levels from days 1 to 30 after CHIMERA in rats, yet also saw no rotarod impairment one month after injury [57]. However, they did not investigate acute rotarod deficits, and their injury parameters differed slightly. This suggests that our current mTBI paradigm may induce transient motor deficits specific to tasks requiring balance and co-ordination without impeding motor function to a degree which limits general activity.

A study by Edem and colleagues showed that ketamine could rescue sensorimotor deficits concurrent with attenuating inflammation after chronic unpredictable mild stress [58]. As our data during the ketamine infusion show, ketamine can certainly influence motor behavior. Though we did not see it rescue rotarod deficits, future studies should investigate this important relationship between ketamine and motor outcomes.

The findings from the current study show that this injury did not result in deficits in either ASR or PPI. This is important because CHIMERA has been shown to affect sensory function and can result in visual deficits [59]. To our knowledge, this is the first study showing that CHIMERA-injured animals have normal hearing after injury. CHIMERA has been shown to result in some behavioral deficits associated with psychiatric disorders, including anxiety, depression, and enhanced fear memory [55]. As PPI (a measure of sensory gating) deficits have been implicated in some psychiatric disorders, particularly related to fear behaviors [60], it is important to note that we did not see any significant differences between the sham and injured animals.

It is clinically significant that we did not see any significant adverse effects of ketamine on ASR and PPI outcomes. Ketamine is a potent neuropsychiatric drug with effects that can endure after it is metabolized. Therefore, its use in TBI, which is comorbid with many neuropsychiatric disorders, should be approached with caution. When given immediately before PPI testing, ketamine has been shown to further reduce PPI in blast-injured animals [61]. However, our findings demonstrate that ketamine does not show this effect in mTBI animals when given several days before testing. The timing of ketamine administration is, therefore, critical to consider when investigating the effects that it has on neuropsychiatric outcomes after mTBI. Additionally, ketamine’s effects on neuropsychiatric disorders have been postulated to be associated with its effects on inflammation [62,63]. As our findings show that ketamine significantly affects cytokines after administration, the interaction between ketamine’s effects on inflammation and neuropsychiatric outcomes warrants further study.

The CHIMERA injury did not produce significant changes in microglial or astrocyte activation in the selected brain areas. A summary of CHIMERA findings by McNamara and colleagues described several studies in which glial activation and proliferation were seen in multiple brain areas, most consistently the corpus callosum and the optic tract [55]. However, findings differed across studies, likely fluctuating due to different injury parameters (impact intensity, impact number, and time point after injury). Additionally, we expected glial changes four days after TBI [12], but different time points should be considered for examination. As our study aimed to produce a mild TBI, it is not surprising to see non-significant inflammatory changes in the brain, which further demonstrates that this injury paradigm produced a very mild injury.

The mTBI can induce complex pathology in axons and myelin [64]. High-velocity head displacement, such as that produced by the CHIMERA model, consistently leads to diffuse axonal injury due to the rotational acceleration forces [55]. As the review by McNamara and colleagues describes, axonal damage after CHIMERA is shown most commonly by silver stain in numerous brain areas and time points after injury [55]. Our data show that, in the selected brain areas, only the optic tract showed significant damage following CHIMERA. The optic tract seems to be one of the most consistent markers of pathological damage in CHIMERA [55] and corresponds with deficits in visual signaling and function [59,65]. It is, therefore, unsurprising that significant axonal damage in this mTBI paradigm is revealed in this particular brain region, as it seems to be more susceptible to damage after CHIMERA. Future studies should use axonal damage in the optic tract as validation of damage in this model. In this study, 10 mg/kg ketamine did not reverse the optic tract damage. This may be due to axonal damage resulting from the biomechanics of the primary injury and not a secondary injury consequence due to inflammation. Injury due to primary mechanical damage may initiate degeneration of the axon, and certain aspects of this axonal degeneration may be irreversible [64]. Therefore, a therapeutic such as ketamine that is intended to treat some secondary rather than primary injury processes may not have dramatic effects on primary axonal pathology [10]. Future studies should look at ketamine use in conjunction with other therapeutics to determine whether it could be beneficial in this aspect of mTBI.

There are several limitations of this study. Not all cytokines were detectable in all plasma samples, and this persisted due to the mild nature of the injury, which did not drastically increase peripheral cytokine levels. We only investigated gross changes in glial proliferation in the current study, so it is possible that examining morphology or quantifying cell counts may have yielded different results. Though we expected mild behavioral deficits due to the nature of mTBI, this also necessitates a greater number of animals due to small effect sizes, making it difficult to assess the effect of ketamine on functional outcomes. Despite this limitation, it is clear from these data that ketamine also had no detectable adverse effects on outcomes, so future studies are necessary to continue to pursue this line of investigation. Finally, this study was only performed in adult male rats. Males and females show different responses in TBI [66], and our previous studies show that ketamine has sex-dependent effects as well [47,67]. Additionally, this study only examined acute changes, up to 4 days post-injury. Therefore, future studies should include both sexes and longer time points to ensure improved clinical translation.

In conclusion, this CHIMERA paradigm produced a mild TBI phenotype in adult male rats. A translationally relevant ketamine infusion after an injury did not result in any adverse functional outcomes, and ketamine showed immunomodulatory effects that could be beneficial after mTBI. Future studies should investigate the effect of low-dose ketamine on immunomodulatory action and functional outcomes after TBIs of different severities. Ketamine’s ability to mitigate secondary injury pathways following mTBI, as well as its utility in acute trauma, makes it an attractive candidate for an evidence-based first-line treatment for mTBI and a potential means to improve patient outcomes worldwide.

## 5. Transparency, Rigor, and Reproducibility Summary

The sample size of rats was 12+ per group based on a power analysis, with extra rats to accommodate catheter patency loss and heterogeneity of injury. A total of 43 rats underwent the sham procedure, 69 rats were subjected to experimental injury, 2 died after injury, and rats were randomly assigned to groups after injury. All rats received either the assigned saline or ketamine dose and completed behavioral testing. Blood was unable to be drawn from 13 rats due to catheter patency loss. Investigators who administered the therapeutic and performed behavioral and histological outcome measures were blinded to group. Therapeutic intervention occurred 1 h after injury. Ketamine came from the same batch of drugs and was diluted in saline the morning of the experiment.

## Figures and Tables

**Figure 1 bioengineering-10-00941-f001:**
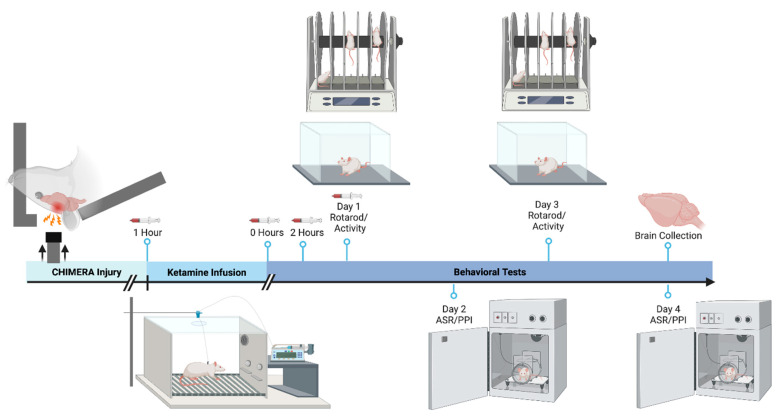
Schematic for experimental timeline. Animals received a two-hour ketamine or saline infusion one hour after a sham or CHIMERA procedure. Blood draws were performed 1, 3, 5, and 24 h after injury (prior to, 0, 2, and 24 h after the ketamine infusion). Rotarod and locomotor activity testing were conducted on days 1 and 3 following injury, and ASR/PPI testing was conducted on days 2 and 4. Animals were euthanized and their brains were collected after behavioral testing on day 4. Figure created with BioRender.com (accessed on 15 January 2023).

**Figure 2 bioengineering-10-00941-f002:**
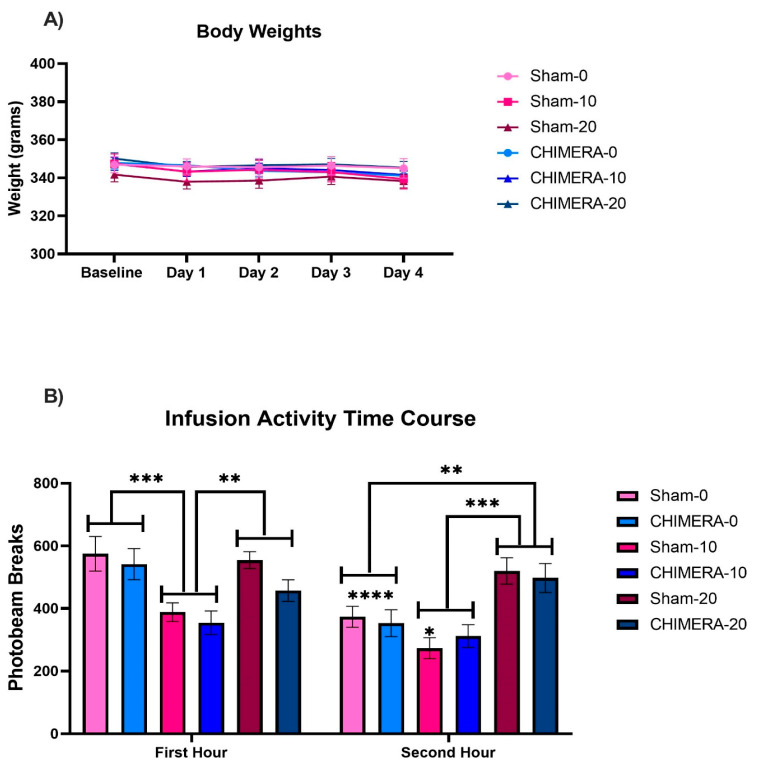
(**A**) There were no differences in body weights between the groups based on either CHIMERA or ketamine condition at any of the time points. (**B**) Ketamine significantly affected spontaneous locomotor activity levels during the infusion. There were no significant effects of CHIMERA. Solo stars represent significant comparisons between doses across the first and second hour, and stars with bars represent significant ketamine dose comparisons within each hour. * *p* < 0.05, ** *p* < 0.01, *** *p* < 0.001, and **** *p* < 0.0001.

**Figure 3 bioengineering-10-00941-f003:**
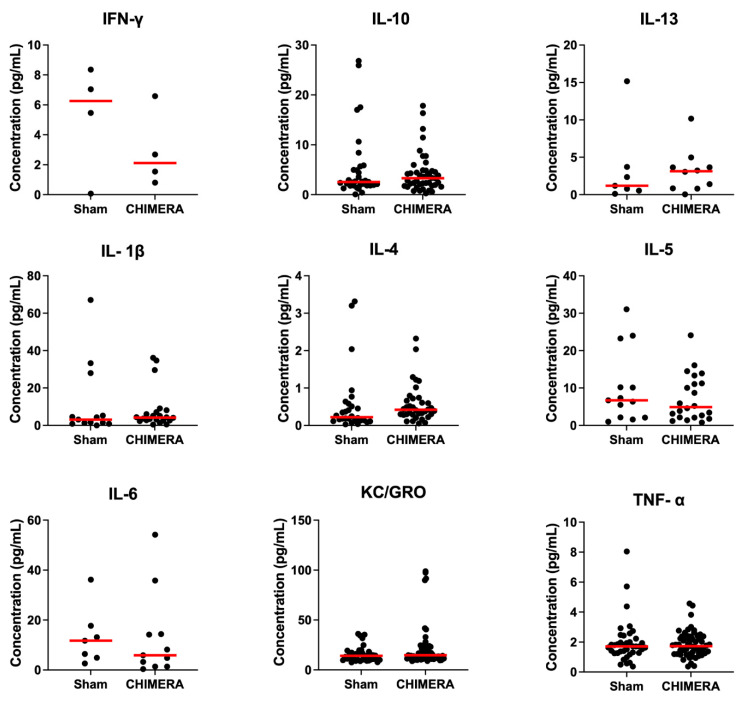
Cytokine expression in plasma does not change one hour after CHIMERA injury. Data are presented as individual subjects, with median bars in red. Varying number of points across cytokines and sham/CHIMERA groups are due to some cytokines being undetectable in samples. KC/GRO and TNF-α were detectable in all samples (sham: n = 39, CHIMERA: n = 58). Unpaired *t*-tests showed no significant differences between sham and CHIMERA-injured animals at this time point. *p*-values are as follows: IFN-γ (*p* = 0.336), IL-10 (*p* = 0.403), IL-13 (*p* = 0.912), IL-1β (*p* = 0.483), IL-4 (*p* = 0.971), IL-5 (*p* = 0.31), IL-6 (*p* = 0.983), KC/GRO (*p* = 0.075), and TNF-α (*p* = 0.555).

**Figure 4 bioengineering-10-00941-f004:**
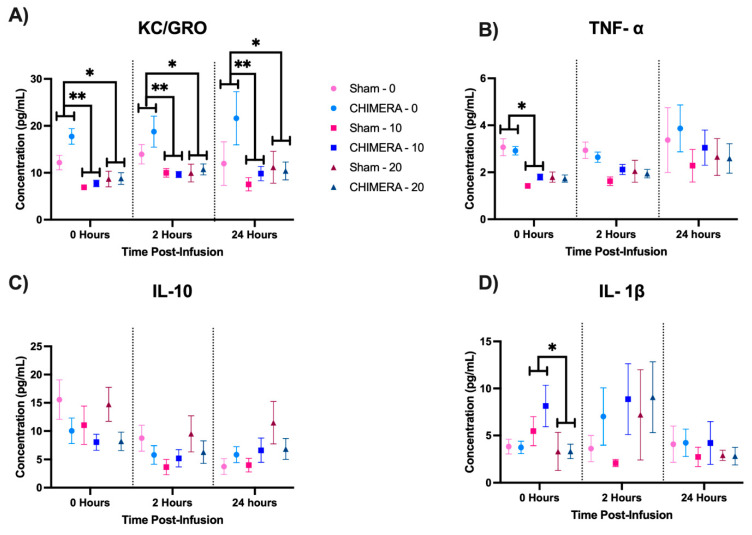
Ketamine significantly modulates several cytokines after an IV infusion. There was no significant effect of CHIMERA. Post hoc comparisons show differences between doses at each time point. (**A**) Both 10 and 20 mg/kg ketamine doses reduce KC/GRO concentration compared to saline at each time point. (**B**) The 10 mg/kg ketamine dose reduces TNF-α 0 h after the infusion. (**C**) IL-10 showed a significant main effect of time. (**D**) IL-1β is significantly reduced at 0 h in rats that received 20 mg/kg ketamine compared to rats that received 10 mg/kg ketamine. * *p* < 0.05, ** *p* < 0.01. Due to catheter patency loss, sample sizes for blood data are as follows: Sham-0 (n = 13), Sham-10 (n = 13), Sham-20 (n = 13), CHIMERA-0 (n = 22), CHIMERA-10 (n = 19), and CHIMERA-20 (n = 17).

**Figure 5 bioengineering-10-00941-f005:**
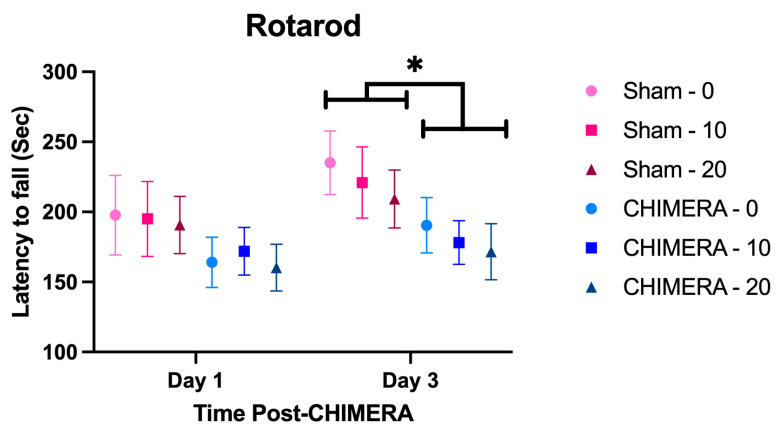
CHIMERA-injured animals show deficits on the rotarod on day 3 after injury. Data are shown as the mean ± SEM. There were no differences between the groups at baseline. There was a significant main effect of CHIMERA (*p* = 0.025) and of time (*p* < 0.001). Post hoc tests reveal that there were no significant differences between the groups either on baseline or day 1 after injury. However, on day 3, CHIMERA-injured animals showed reduced latency to fall off the rotarod as compared to Sham animals (*p* = 0.013), suggesting motor deficits due to CHIMERA. * *p* < 0.05.

**Figure 6 bioengineering-10-00941-f006:**
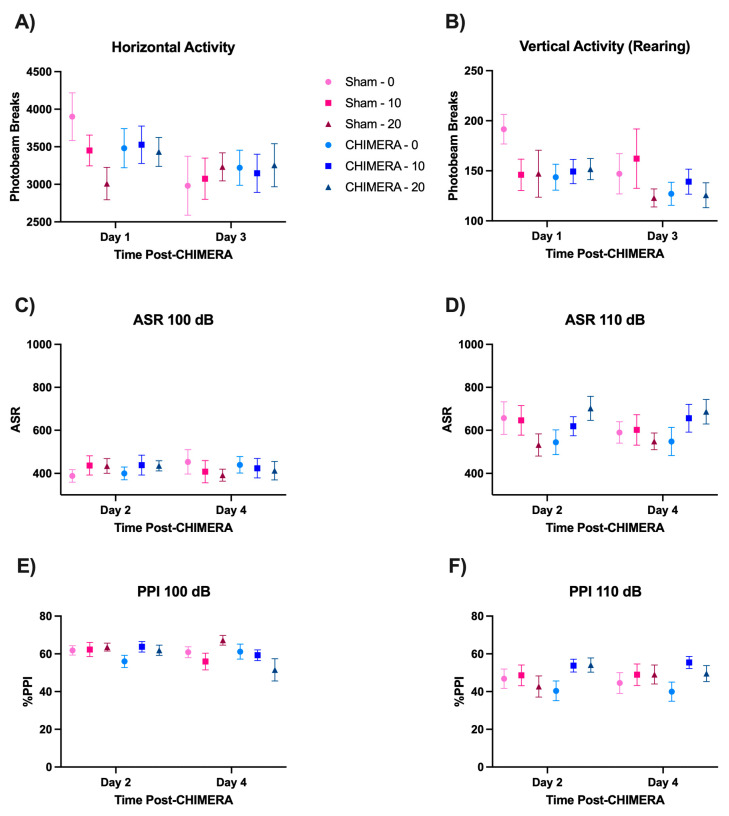
CHIMERA and ketamine induce no significant behavioral deficits. There were no baseline differences between groups prior to CHIMERA and ketamine infusion. (**A**) Horizontal activity on days 1 and 3 after CHIMERA. (**B**) Vertical activity (rearing) on days 1 and 3 after CHIMERA. (**C**) ASR 100 dB on days 2 and 4 after CHIMERA. (**D**) ASR 110 dB on days 2 and 4 after CHIMERA. (**E**) PPI at 100 dB on days 2 and 4 after CHIMERA. (**F**) PPI at 110 dB on days 2 and 4 after CHIMERA.

**Figure 7 bioengineering-10-00941-f007:**
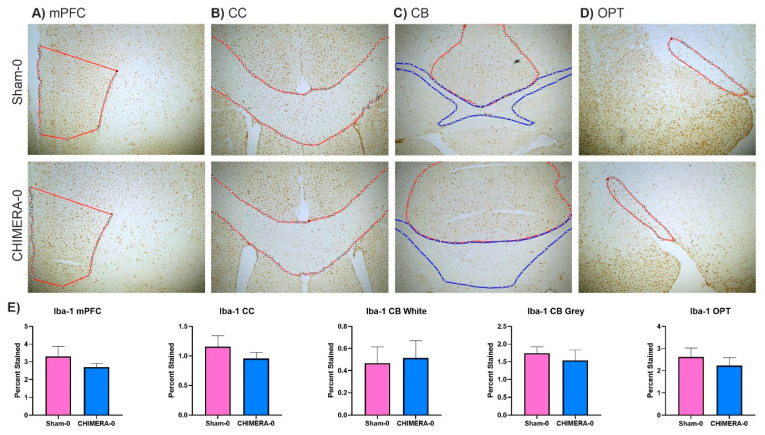
CHIMERA does not induce microglial proliferation in the selected brain areas four days after injury. Representative images from each brain area at 4× magnification. ROIs drawn for percent staining analysis are drawn in red. In the CB grey matter is drawn in red, white matter in blue. For mPFC and OPT, right and left sides were averaged for an overall score. (**A**) Medial prefrontal cortex (mPFC). (**B**) Corpus callosum (CC). (**C**) Cerebellum (CB). (**D**) Optic tract (OPT). (**E**) Quantification of each percent-stained area. There were no significant differences between the sham and CHIMERA groups. mPFC (*p* = 0.33), CC (*p* = 0.484), OPT (*p* = 0.486), CB White (*p* = 0.832), and CB Grey (*p* = 0.56).

**Figure 8 bioengineering-10-00941-f008:**
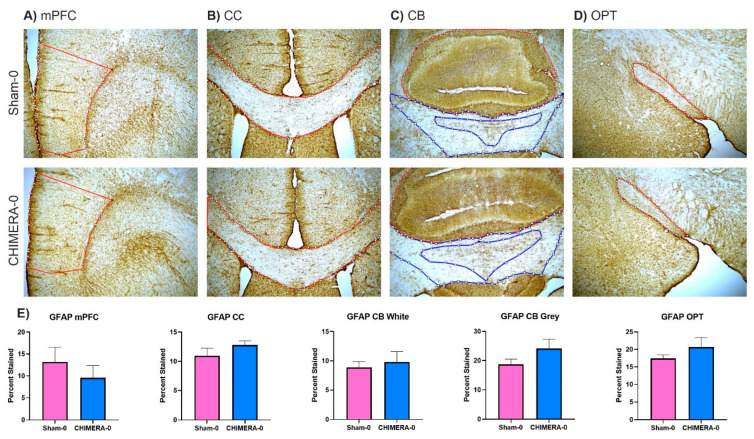
CHIMERA does not induce astrocytic proliferation in the selected brain areas four days after injury. Representative images from each brain area at 4× magnification. ROIs drawn for percent staining analysis are drawn in red. In the CB grey matter is drawn in red, white matter in blue. For mPFC and OPT, right and left sides were averaged for an overall score. (**A**) Medial prefrontal cortex (mPFC). (**B**) Corpus callosum (CC). (**C**) Cerebellum (CB). (**D**) Optic tract (OPT). (**E**) Quantification of each percent-stained area. There were no significant differences between the sham or CHIMERA groups in unpaired *t*-tests. mPFC (*p* = 0.438), CC (*p* = 0.239), OPT (*p* = 0.243), CB White (*p* = 0.644), and CB Grey (*p* = 0.157).

**Figure 9 bioengineering-10-00941-f009:**
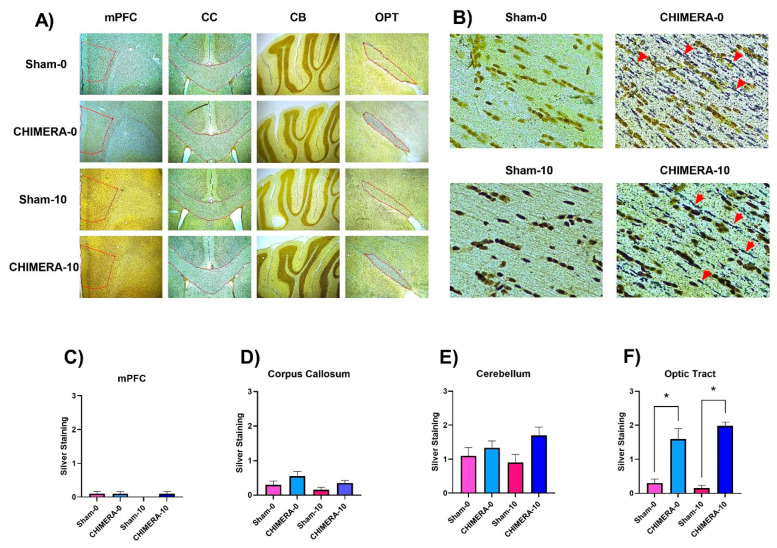
CHIMERA induces significant axonal damage in the optic tract four days after injury. (**A**) Representative images of evidence of axonal damage shown through silver stain uptake at 4× magnification for each brain section and group. Areas examined are outlined in red for clarity. CB was evaluated as a whole. For mPFC and OPT, the right and left sides were averaged for an overall score. (**B**) Representative images of optic tract staining for each group at 40× magnification using sections selected in (**A**). Red arrowheads indicate degenerating nerve fibers. (**C**–**F**) shows quantitative scores of staining in each brain area. (**C**) Medial prefrontal cortex (mPFC) showed no difference between the groups (*p* = 0.514). (**D**) Corpus callosum (CC) showed no difference between the groups (*p* = 0.113). (**E**) Cerebellum (CB) showed no significant differences between the groups (*p* = 0.109). (**F**) Optic tract (OPT) showed significantly more silver uptake in CHIMERA groups (*p* < 0.0001). Multiple comparisons showed significant differences between the Sham-0 and CHIMERA-0 groups (*p* = 0.03), the Sham-0 and CHIMERA-10 groups (*p* = 0.004), the CHIMERA-0 and Sham-10 groups (*p* = 0.003), and the Sham-10 and CHIMERA-10 groups (*p* = 0.0003). * *p* < 0.05.

## Data Availability

Data available upon request to a corresponding author.

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
