# Peer review of "Effects of a Subanesthetic Ketamine Infusion on Inflammatory and Behavioral Outcomes after Closed Head Injury in Rats"

_bioengineering, 2023, doi:10.3390/bioengineering10080941_

Round 1
Reviewer 1 Report
"Effects of a Subanesthetic Ketamine Infusion on Inflammatory 2 and Behavioral Outcomes after Closed Head Injury in Rats"
Unfortunately, it is often underestimated that even comparatively minor brain damage can lead to traumatic axonal tears. In imaging procedures such as CT or MRI, the brain structures appear completely intact; however, on a microscopic level, many neuronal connections have lost contact. This leads to a lack of resilience in patients with slight but persistent deficits in concentration, memory and perception - which often nobody believes, since no damage can be seen on the MRI. In addition, this minimal but very extensive damage in the brain then leads to inflammatory processes and edema, which cause further damage. From this point of view, the present study is very important, it tries to find protective factors that reduce the extent of the damage consequences.
The manuscript examines the effects of ketamine on mild traumatic brain injury. Ketamine is an anesthetic that is commonly used, especially in children and animals. In contrast to other anesthetics, it has circulatory-stabilizing and anti-inflammatory effects. On the downside, it causes severe hallucinations and particularly a feeling of dissociation from the body. It is therefore also used by drug junkies and to induce near-death experiences with the feeling of leaving the body. Because of the hallucinations, it is given to adults only after accidents.
The authors correctly write that there are hardly any studies examining whether ketamine could have a protective effect on brain damage due to its positive properties. However, only mild craniocerebral trauma was examined in the present study. However, it would probably also have been difficult to test different degrees of severity of a craniocerebral trauma in addition to various other variables.
The mild traumatic brain injury using a Closed-Head Impact Model of Engineered Rotational Accelera (CHIMERA) apparently only resulted in damage to the optic nerve. Maybe it was too mild? Ultimately, only relatively small differences were found between the group with mild brain damage and the rats with a placebo treatment. On the other hand, two rats died after brain damage; so it can't have been that easy after all.
The study is an animal experiment, which is probably inevitable before any human studies are conducted. The ketamine infusion was given 1 hour after the injury, which is probably close to the treatment of an actual brain-injured person. The dose was varied (none, 10 and 20 mg/kg). That's quite a lot, as far as I know, the ketamine administration in patients is only a few milligrams per kilogram of body weight. As a drug, 2 mg per kg is considered sufficient for psychedelic experience in humans.
Unfortunately, the behavioral tests were only carried out on the first four days; it remains questionable whether there would have been a protective long-term effect?
As a placebo, a saline infusion was administered instead of ketamine.
In addition to the behavioral examination, immune parameters were checked and later, microscopically, brain regions were examined for staining included the medial prefrontal cortex, the corpus callosum, the cerebellum, and the optic tract.
During the administration of the medication, the anesthetic effects logically led to a reduction in physical activity. However, ketamine administration had no long-term effects on the behavior of the rats. A reduction in immune functions could be demonstrated. Unfortunately, it remains a mystery whether ketamine administration really has a protective effect.
All in all, a very well-rounded study, in which many aspects were well taken into account, in addition to behavior, also immunological parameters and changes at cell level. From a scientific point of view, the experiment seems very good to me. The extensive illustrations and the brain sections are convincing. Overall a good article for which I cannot make any serious suggestions for improvement. From my point of view, the authors are on an interesting path here.
Author Response
Unfortunately, it is often underestimated that even comparatively minor brain damage can lead to traumatic axonal tears. In imaging procedures such as CT or MRI, the brain structures appear completely intact; however, on a microscopic level, many neuronal connections have lost contact. This leads to a lack of resilience in patients with slight but persistent deficits in concentration, memory and perception - which often nobody believes, since no damage can be seen on the MRI. In addition, this minimal but very extensive damage in the brain then leads to inflammatory processes and edema, which cause further damage. From this point of view, the present study is very important, it tries to find protective factors that reduce the extent of the damage consequences.
The manuscript examines the effects of ketamine on mild traumatic brain injury. Ketamine is an anesthetic that is commonly used, especially in children and animals. In contrast to other anesthetics, it has circulatory-stabilizing and anti-inflammatory effects. On the downside, it causes severe hallucinations and particularly a feeling of dissociation from the body. It is therefore also used by drug junkies and to induce near-death experiences with the feeling of leaving the body. Because of the hallucinations, it is given to adults only after accidents.
The authors correctly write that there are hardly any studies examining whether ketamine could have a protective effect on brain damage due to its positive properties. However, only mild craniocerebral trauma was examined in the present study. However, it would probably also have been difficult to test different degrees of severity of a craniocerebral trauma in addition to various other variables.
The mild traumatic brain injury using a Closed-Head Impact Model of Engineered Rotational Accelera (CHIMERA) apparently only resulted in damage to the optic nerve. Maybe it was too mild? Ultimately, only relatively small differences were found between the group with mild brain damage and the rats with a placebo treatment. On the other hand, two rats died after brain damage; so it can't have been that easy after all.
- Thank you for your comments. This model was intended to produce a mild TBI, so we did not expect to see extensive brain damage after the injury. As the reviewer mentioned above as well, it would have been difficult to test different degrees of severity of trauma, though that would be important for future studies. The CHIMERA paradigm chosen for this study has been shown to produce an injury similar to a concussion injury in humans (Namjoshi, Cheng et al. 2017), and clinically, many patients do not develop significant symptoms after mild TBI (Sigurdardottir, Andelic et al. 2014). Mild TBI is not well defined in preclinical studies and the range in injury severity seen is inconsistent across animal studies, (Shultz, McDonald et al. 2017). Thus we present this model as a clinically relevant mild TBI model that does not lead to widespread neuropathology or behavioral deficits.
The study is an animal experiment, which is probably inevitable before any human studies are conducted. The ketamine infusion was given 1 hour after the injury, which is probably close to the treatment of an actual brain-injured person. The dose was varied (none, 10 and 20 mg/kg). That's quite a lot, as far as I know, the ketamine administration in patients is only a few milligrams per kilogram of body weight. As a drug, 2 mg per kg is considered sufficient for psychedelic experience in humans.
- Rodents have much faster metabolisms than humans, so it is common practice that animals are administered with higher doses of drug in order to achieve the same effects. This is why our lab focuses on “clinically-relevant” doses of ketamine, and validate the doses chosen by investigating dissociative and analgesic effects after administration in order to characterize the dose strength in rodents. We previously published studies on the pharmacokinetics and pharmacodynamics of intravenous ketamine infusions in Sprague Dawley rats that have demonstrated higher mg/kg dosing to achieve analgesia without dissociative effects in rats (Radford, Park et al. 2017, Radford, Park et al. 2018, Radford, Berman et al. 2022).
Unfortunately, the behavioral tests were only carried out on the first four days; it remains questionable whether there would have been a protective long-term effect?
- The authors agree that we only investigated acute behavioral effects in this study, and hope to investigate longer time points in future studies as the protective long-term effects are very important.
As a placebo, a saline infusion was administered instead of ketamine.
In addition to the behavioral examination, immune parameters were checked and later, microscopically, brain regions were examined for staining included the medial prefrontal cortex, the corpus callosum, the cerebellum, and the optic tract.
During the administration of the medication, the anesthetic effects logically led to a reduction in physical activity. However, ketamine administration had no long-term effects on the behavior of the rats. A reduction in immune functions could be demonstrated. Unfortunately, it remains a mystery whether ketamine administration really has a protective effect.
- We agree with this conclusion, and plan to pursue further studies in order to investigate ketamine’s protective effects post-TBI.
All in all, a very well-rounded study, in which many aspects were well taken into account, in addition to behavior, also immunological parameters and changes at cell level. From a scientific point of view, the experiment seems very good to me. The extensive illustrations and the brain sections are convincing. Overall a good article for which I cannot make any serious suggestions for improvement. From my point of view, the authors are on an interesting path here.
- Once again, we thank you for your comments on this study.
Reviewer 2 Report
The manuscript is well-designed and complicated for neurological parameters. However, before the acceptance of the manuscript discussion section should be shortened and limitations of the research should be indicated.
Author Response
The manuscript is well-designed and complicated for neurological parameters. However, before the acceptance of the manuscript discussion section should be shortened and limitations of the research should be indicated.
- Thank you for your comments. Per your suggestions, the discussion section has been edited and shortened, and the limitations paragraph at the end of the discussion has been extended.